# Dynamic Skill Adaptation for Large Language Models

## Abstract

We present Dynamic Skill Adaptation (DSA), an adaptive and dynamic framework to adapt novel and complex skills to Large Language Models (LLMs). Compared with previous work which learns from human-curated and static data in random orders, we propose to first automatically generate and organize the training data by mimicking the learning pathways of human and then dynamically tailor the training data based on the training dynamics. Specifically, inspired by the learning structures and teaching strategies in the human education system, we first construct a skill graph by decomposing complex skills into sub-skills and arranging them based on their dependencies in human syllables. For every skill, we utilize LLMs to generate both textbook-like data which contains detailed descriptions of skills for pre-training and exercise-like data which targets at explicitly utilizing the skills to solve problems for instruction-tuning. Furthermore, during the instruction-tuning, we dynamically update the training data which down-weight easy-to-learn examples, generate more complex examples, and filter out data with errors. Experiments on large language models such as LLAMA and Mistral demonstrate the effectiveness of our proposed methods in adapting math reasoning skills and social study skills.

## 1 Introduction

Large Language Models (LLMs) have witnessed a significant rise in popularity and utility across various domains in NLP such as text generation, machine translation, and question-answering systems (Brown et al., 2020; Radford et al., 2019; Smith et al., 2022; Chowdhery et al., 2022; Lewkowycz et al., 2022; Sanh et al., 2021; Wei et al., 2021; Mishra et al., 2022; Chung et al., 2022; Ouyang et al., 2022; OpenAI, 2023; Touvron et al., 2023). The success of LLMs such as ChatGPT and GPT-4 and its predecessors has demonstrated their ability to learn, understand, and generate human-like text based on massive amounts of existing data(Qin et al., 2023; Ziems et al., 2024; OpenAI, 2023; Wang et al., 2022; Dubois et al., 2024). Despite their remarkable achievements in general benchmarks and tasks, these current LLMs often fail when it comes to specialized domains which require complex and novel skills such as math reasoning, coding, and etc. (Khot et al., 2022; Chen et al., 2023; Dziri et al., 2023; Xu et al., 2023; Shao et al., 2024; Frieder et al., 2024).

When adapting specific and complex skills to LLMs that are pre-trained on general corpus, there are several challenges. First, LLMs may lack specific domain knowledge that is necessary to understand and generate content in a specialized field such as math. Adapting to them requires mechanisms to incorporate domain-specific terminology, concepts, and context. However, specialized, or complex skills often only have limited data available for fine-tuning. While previous approaches mainly collect existing data Yue et al. (2023) or generate synthetic data from a small set of human-written seed examples Wang et al. (2022); Xu et al. (2023) and mix them together to fine-tune the models Xie et al. (2024); Yue et al. (2023); Li et al. (2024), it is still under-explored how to select, organize, and utilize domain-specific data to effectively learn novel and complex skills Chen et al. (2024). Furthermore, LLMs are often easy to overfit limited and static training data during fine-tuning Xu et al. (2023); Shao et al. (2024), which could result in sub-optimal performance.

To overcome these challenges, we draw inspirations from teaching strategies in human education systems (Weinstein & Mayer, 1983). Good teaching strategies often involve several key steps: (i) Teachers would decompose and **organize** content into several levels, starting from easier childhood education up to higher education and beyond (Brighouse, 2006). (ii) Within each level, teachers would

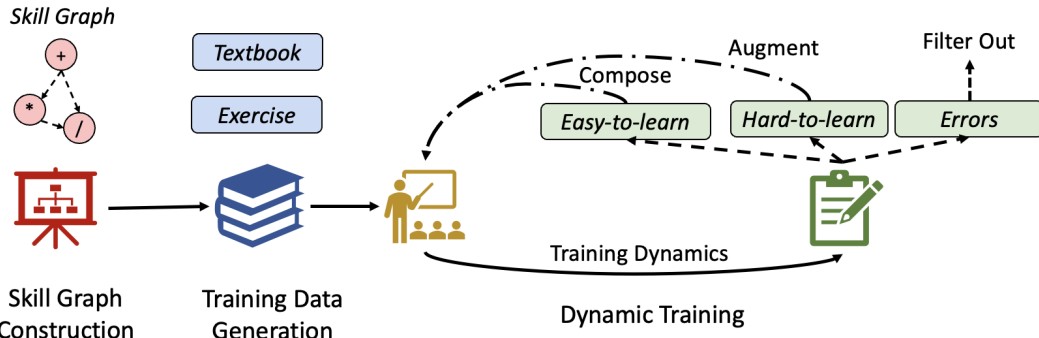

Figure 1: The overall process of our Dynamic Skill Adaptation framework. For a given complex skills, we first built the skill graph where sub-skills are organized following their dependencies (e.g., mastering summing first and then learning multiplying). Then we generate textbook-like descriptions for every skill and generate exercise-data where the skills that have been learned need to be explicitly used to solve the generated problems. During the training, we dynamically adjust the training data based on the training dynamic.

**rehearse** previous knowledge and link them to more complex content with detailed **elaborations**. (iii) During the entire process, teachers would actively **monitor** the learning of students and dynamically adjust the teaching materials. We believe that a learning framework that follows and utilizes human learning strategies and structures would have the potential to allow LLMs to better adapt complex skills.

To this end, we introduce Dynamic Skill Adaptation (DSA), a framework specifically designed for LLMs that adaptively generates and organizes training data automatically and allows LLMs to acquire specialized or novel skills dynamically. Specifically, inspired by the **organization** strategy in teaching (Weinstein & Mayer, 1983), we first build a skill graph based on human learning syllables which decompose complex skills such as calculus into sub-skills and further arrange them based on their dependencies so that model could learn prerequisite knowledge and then higher-level knowledge. Next, following the **elaboration** and **rehearsal** strategy, we automatically generate detailed textbook-like descriptions for each skill using LLMs like GPT4 as well as the exercise data where the skills that have been learned need to be explicitly used to solve the generated problems. In addition, with the **monitoring** strategy, during the training, we would dynamically adjust the training data based on the learning dynamics where we generate more complex and hard-to-learn examples, filter out data with errors and down-weight easy-to-learn examples. Experiments on large language models such as LLAMA (Touvron et al., 2023) and Mistral (Jiang et al., 2023) demonstrate the effectiveness of our proposed methods in adapting math reasoning skills and social study skills.

Our work has three major contributions: (i) We propose to generate and organize the training corpus that contains both textbook and exercise-like data based on skill graphs for LLMs to adapt novel skills inspired by human teaching strategies. (ii) We introduce the dynamic training mechanism that adjusts the training data based on the training dynamics to avoid overfitting static data. (iii) Experiments on several LLMs in Math and Social Study domain and extensive ablation studies demonstrate the effectiveness of our introduced Dynamic Skill Adaptation framework.

## 2 RELATED WORK

**Large Language Model**    Large language models have witnessed extensive progress recently (Brown et al., 2020; Radford et al., 2019; Smith et al., 2022; Chowdhery et al., 2022; Lewkowycz et al., 2022; Sanh et al., 2021; Wei et al., 2021; Mishra et al., 2022; Chung et al., 2022; Ouyang et al., 2022; OpenAI, 2023; Touvron et al., 2023; Wei et al., 2022) and have shown superior performance in a wide range of general tasks such as natural language understanding (Hendrycks et al., 2021; Qin et al., 2023), reasoning (OpenAI, 2023) and code generation (Touvron et al., 2023). However, LLMs that is

pre-trained on general data usually fail domain specific tasks (Xu et al., 2023; Shao et al., 2024; Shah et al., 2022; Cheng et al., 2023) or tasks that require complex skills (Yue et al., 2023; Xu et al., 2023; Zhu et al., 2023; Chen et al., 2023; Khot et al., 2022). Instead of general LLMs, in this work, we focus on how to adapt pre-trained general LLMs automatically and efficiently with specialized and complex skills.

**Large Language Model for Specialized Domain** Recent approaches have also explored generating or collecting domain-specific data for tuning models for specialized domain such as math (Shao et al., 2024; Yue et al., 2023; Xu et al., 2023) and coding (Nijkamp et al., 2022; Luo et al., 2023; Li et al., 2023). They either collect a wide range of online data (Wang et al., 2023; Li et al., 2023; Shao et al., 2024) or generate instruction tuning data with LLMs (Li et al., 2023; Yue et al., 2023; Xu et al., 2023; Luo et al., 2023; Toshniwal et al., 2024) through techniques such as Self-instruct(Wang et al., 2022) or Evol-instruct (Xu et al., 2023; Luo et al., 2023). However, these methods usually randomly mix all the data together while ignoring the dependencies and relations between different data (Xie et al., 2024; Chen et al., 2024). Also, the diversity is often restricted by seed instructions or seed topics. As a result, the training process might suffer from overfitting on these static data. To overcome these issues, we propose to not only generate the training data through LLMs but also organize them following human learning orders and dynamically update them during adapting the LLMs to novel and complex skills for specialized domains.

**Curriculum Learning** Curriculum learning (Bengio et al., 2009) propose to train the model with data that is arranged from easy samples to hard ones with designed pacing functions and mixing rates (Soviany et al., 2022; Wang et al., 2021; Portelas et al., 2020; Matiisen et al., 2019; Jiang et al., 2015) through assigning learnability scores (Xu et al., 2020; Lu et al., 2024; Bejan et al., 2023) or utilizing agents to generate harder examples (Feng et al., 2023; Fan & Jaggi, 2023; Balloccu et al., 2024). Saxena et al.; Mindermann et al. also explores parametrizing and ordering samples with importance. Chen et al. propose algorithms to select the orders of data from different tasks by enumerating all the sequences and selecting the best sequence based on the performances on smaller scale experiments. While our work is inspired by curriculum learning, we focus more on the skill-level: instead of ranking specific examples, we model the order of different skills based on their dependencies and does not necessarily follow an easy-to-hard manner.

# 3 METHODS

The first step towards equipping LLMs with domain-specific knowledge efficiently and adaptively is to review how humans learn new skills in a new domain comprehensively. Good teaching includes teaching students how to learn, remember, think, and motivate themselves through organization, rehearsal and elaboration, comprehension monitoring(Weinstein & Mayer, 1983). Motivated by these human learning strategies, we propose the Dynamic Skill Adaptation (DSA) framework for LLM shows in Figure 1 and Algorithm 1, which consists of several key components: Skill Graph Construction (Section 3.1), Training Data Generation (Section 3.2) and Dynamic Training (Section 3.3).

## 3.1 SKILL GRAPH CONSTRUCTION

When human learn new skills, it is necessary for teachers to structure and arrange the information to make it more understandable and easier to remember for students, such as creating outlines, mind maps, charts, or using other organizational tools to group related concepts together(Weinstein & Mayer, 1983). As a result, one key component of DSA involves constructing a skill graph and organizing them following the learning structures. For example, to learn skills about calculus in math, models need to first learn skills such as algebra, function, geometry, trigonometry, etc. Every node in our skill graphs is a specific skill, and the edges between them represent their dependence.

In practice, when adapting a complex skill $S$ (e.g., *calculus* in math), we build the skill graphs in two ways: (i) we gather the basic skills in human learning syllabus [1] and the edges are pointed from lower-level skills to higher-level skills; (ii) we recursively prompt GPT4 (e.g., *what are the basic skills that are required to learn Calculus*) to decompose complex skills into sub-skills and the edges

---

[1]For example, https://www.ixl.com/

are pointed from sub-skills to complex skills. We then merge the skill graphs from both human syllabus and LLM generations into a final skill graph $G$. Example sub-graphs in our skill graphs are visualized in Figure 3 and Figure 4. Following the skill graphs we constructed, we will organize the training data and train the models to acquire skills from lower-levels to higher-levels based on the skill graphs (i.e., learn lower-level knowledge first before learning higher-level knowledge).

## 3.2 TRAINING DATA GENERATION

In human learning systems, elaboration goes beyond rote memorization and involves expanding the material, making connections to previous knowledge, and deepening understanding. Rehearsal is the process of repeatedly going over information to commit it to memory. This might involve reading notes, rephrasing ideas, or reciting key facts. It helps in retaining information in memory and can be useful for rote memorization. Based on these human learning strategies (Weinstein & Mayer, 1983), in our DSA, we then automatically generate textbook data for elaboration and exercise data for rehearsal to learn new skills. During the training, we would first pre-train LLMs with the textbook descriptions following the orders in the skill graph we constructed in Section 3.1 and then instruction-tune LLMs with the exercise data.

**Textbook Generation** For every node $s$ in the generated skill graph $G$, we instruct GPT4 to generate textbook descriptions (Li et al., 2023) which could be used for pre-training. Specifically, we regularize the generation of textbook descriptions to (i) link the current skill with its predecessors in the skill graph $G$, (ii) cover as much detail with both descriptions and examples, (iii) highlight all the key concepts at the end of the descriptions, and (iv) provide homework that covers every key concept for the current skill.

**Exercise Generation** For the nodes in $G$, we further instruct GPT4 to generate exercise problems which could be used for instructional tuning. We intend to generate questions that each of them would leverage multiple skills in $G$ which are different from the homework which only cover one specific skill in the textbook generation stage. As a result, for every generation, we first randomly sample different numbers of skills from the skill graph $G$ and then instruct GPT4 to generate problems which requires the provided skills to solve. When generating answers for the exercise problems, we regularize the generated reasoning steps to be explicitly grounded in specific skills (Chen et al., 2023) and further improve quality through self-consistency (Wang et al., 2022).

## 3.3 DYNAMIC TRAINING

From a human learning perspective, effective learning involves students to actively assess their own understanding of the material. It is about being aware of when students don't fully grasp a concept and taking steps to fill in the gaps (Weinstein & Mayer, 1983). Building upon these insights, we propose a dynamic training scheme to dynamically update the training data based on learning curves. Specifically, after pre-training on textbook-like data, during the instructional tuning of exercise data, we would further categorize and adjust the training data to make models better adapt novel skills.

To distinguish among different types of training examples, inspired by Swayamdipta et al., we utilize two metrics:

- The average loss $\hat{L}_i$ for one example $(x_i, y_i)$ across $E$ epochs:

$$\hat{L}_i = \frac{1}{E} \sum_{e=1}^{E} L\left(y_i, F(x_i)\right)$$

where $F$ is the learned LLM. Intuitively, the lower average loss $\hat{l}_i$ means that the instance is easier for the given LLM.

- The variance for the predictions losses $\hat{\sigma}_i$ one example $(x_i, y_i)$ across $E$ epochs:

$$\hat{\sigma}_i = \sqrt{\frac{\sum_{e=1}^{E} \left(L\left(y_i, F(x_i)\right) - \hat{l}_i\right)^2}{E}}$$

Table 1: Data statistics including the number of skills and the number of textbook and exercise tokens generated from LLMs for training.

| Levels | # of Math Skills | # of tokens | # of Social Study Skills | # of tokens |
|---|---|---|---|---|
| Pre-K | 135 | 103,925 | - | - |
| Kindergarten | 258 | 205,209 | 31 | 22,646 |
| First Grade | 284 | 231,465 | 41 | 29,869 |
| Second Grade | 349 | 291,007 | 62 | 47,238 |
| Third Grade | 425 | 370,830 | 98 | 80,001 |
| Fourth Grade | 415 | 372,806 | 111 | 97,794 |
| Fifth Grade | 447 | 405,265 | 111 | 97,551 |
| Sixth Grade | 413 | 378,255 | 150 | 136,589 |
| Seventh Grade | 374 | 341,207 | 205 | 180,191 |
| Eighth Grade | 392 | 361,625 | 181 | 168,095 |
| Algebra 1 | 396 | 388,712 | - | - |
| Algebra 2 | 367 | 384,239 | - | - |
| Geometry | 277 | 266,328 | - | - |
| Pre-calculus | 375 | 397,065 | - | - |
| Total | 4,907 | 4,497,938 | 990 | 859,974 |

Intuitively, the lower variance means that LLM $F$ predict the same answers consistently while high variance means that the model is indecisive across training.

Based on these two measures, we would first compute a baseline loss $L_b$ and variance $\sigma_b$ which are the losses and variance after fine-tuning with constructed error examples for three epochs. During actual instructional tuning, after every three epochs of training on the actual training examples which leads to an average training loss $L_{average}$ and average variance $\sigma_{average}$ across all training samples, we would divide them into four categories:

- Data with errors, whose training loss $\hat{L}_i$ is larger than $L_b$: $\hat{L}_i \geq L_b$ and variance is smaller: $\hat{\sigma}_i \leq \sigma_b$.

- Hard-to-learn data, whose training loss $\hat{L}_i$ is higher than average loss but less than $L_b$: $L_b \geq \hat{L}_i \geq L_{average}$, and the variance is larger than baselines but less than average: $\sigma_b \leq \hat{\sigma}_i \leq \sigma_{average}$.

- Easy-to-learn data, whose training loss $\hat{L}_i$ is low and less than $L_b$: $L_b \geq \hat{L}_i$ and $L_{average} \geq \hat{L}_i$, and the variance is low: $\sigma_b \geq \hat{\sigma}_i$ and $\sigma_{average} \geq \hat{\sigma}_i$.

- Ambiguous data, which contains all the other data.

We would filter out all the data with errors. For hard-to-learn examples, we would generate more similar data with the use of GPT4 (Wang et al., 2022; Xu et al., 2023). For easy-to-learn examples, we perform compositional augmentation (Ouyang et al., 2023) where we instruct GPT4 to compose different easy problems together to form harder ones. We keep the ambiguous data unchanged. With the updated training set, we then continue the instructional tuning process.

## 4 EXPERIMENTS

### 4.1 EXPERIMENTS SETUP

To demonstrate the effectiveness of our proposed DSA framework, we perform experiments to adapt the skills of calculus and social studies to LLMs which are two challenging subjects in human education (Duncan, 1960; Jarvis, 2012; Kivunja, 2014; Myers, 2006).

**Data Generation** We first construct the skill graph for calculus and social studies. Specifically, as discussed in Section 3.1, we decompose calculus and social studies with GPT4 and further merge

them with the human-curated syllabus [2] into 14 levels of skills for math (4,907 skills in total) and 9 levels of skills for social studies (990 skills in total), as described in Table 1. Skills in lower levels are pointed to skills in higher levels in the constructed skill graph (e.g., Inside one level, pre-request skills like *counting up to 3* are pointed to more complex ones like *counting up to 10*. Across different levels, skills in Pre-K are pointed to skills in Kindergarten. ). Example skill graphs for calculus and social studies are visualized in Figure 3 and Figure 4.

Next, we prompt the GPT4 model with nucleus sampling (Ravfogel et al., 2023) with temperature $T = 0.5$ and top$-p = 0.95$ to generate the textbook descriptions following the constraints stated in Section 3.2 for every skill in the constructed skill graph [3]. Likewise, we utilize the GPT4 model with nucleus sampling ($T = 0.1$ and top$-p = 0.95$ (we use a lower temperature here to make the problems and answers more accurate.)) to generate both the problems and answers for exercise generation as stated in Section 3.2 [4]. On average, every problem requires 3.8 skills to solve. To improve the quality of the generated answers, we apply self-consistency where we set $k = 3$. The total number of generated tokens for textbook descriptions and exercise is shown in Table 1 where we generate 4,497,938 tokens for adapting calculus and 859,974 tokens for adapting social studies.

During the dynamic training, after categorizing the training data with criteria stated in Section 3.3, we also use GPT4 with nucleus sampling ($T = 1.0$ and top$-p = 0.95$ (we use a higher temperature here to improve the diversity.)) to generate problems which are similar to the given hard-to-learn examples and generate more complex problems by instruct the models to compose two different easy-to-learn problems. Similarly, we apply self-consistency with $k = 3$ to generate the answers for these newly generated problems.

**Evaluation Set**   To evaluate the abilities for calculus, we utilize the Pre-Calculus subset in MATH benchmark (Hendrycks et al., 2021). For social studies, we collect multiple choice exams from online [5], which results in an evaluation set that consists of 1430 questions[6]. In addition, to evaluate the generalization abilities after adapting to specialized domain like calculus, we further evaluate models on GSM8K (Cobbe et al., 2021), MATH(Hendrycks et al., 2021) and a constructed arithmetic task where we define 200 new mathematical operations in the problem description and test the models if they could understand the context to utilize novel math operations [7].

**Backbone Models and Baselines**   We apply our proposed DSA to both LLAMA2-7/13/70b models (Touvron et al., 2023) and Mistral-7b model (Jiang et al., 2023). During the pre-training on textbook descriptions, for every level of skills, we train the models for two epochs with a learning rate of $3e − 4$ with a linear warm-up of 500 steps and we learn following the sequence from lower-level skills to higher-level skills. During the instruction-tuning on exercise data, we train the models for 5 epochs with a learning rate of $3e − 5$. The batch size is set to 16. We update the training set after every epoch of training.

We compare our learned models with several state-of-the-art baselines including ChatGPT(OpenAI, 2023), LLAMA2-7/13/70b(Touvron et al., 2023), Mistral-7b (Jiang et al., 2023), WizardMATH-7/13/70b (Xu et al., 2023), OpenMath-7b(Toshniwal et al., 2024) and DeepSeekMATH-Inst-7b(Shao et al., 2024).

### 4.2 MAIN RESULTS

We apply our DSA framework to adapt the skills of solving calculus problems and social study problems respectively to LLMs including LLAMA2-7/13/70b and Mistral-7b. The results are shown in Table 2. Compared to LLAMA and Mistral models which learn on general corpus, models learned with LLMs generated instructions for specialized domains like WizardMATH perform better. With a larger scale of human-curated corpus, OpenMath and DeepSeekMATH outperform WizardMATH.

---

[2] https://www.ixl.com for calculus and social studies.

[3] An example is shown in Table 6 in the Appendix.

[4] An example is shown in Table 7 in the Appendix.

[5] https://www.helpteaching.com/search/index.htm?keyword=social+studies and https://www.proprofs.com/quiz-school/topic/3rd-grade-social-study

[6] An example is shown in Table 8 in the Appendix.

[7] An example is shown in Table 9 in the Appendix.

Table 2: Accuracy on Pre-Calculus and Social Studies evaluation sets. We compare our DSA with close-sourced models including ChatGPT and GPT4 and open-sourced models including LLAMA2, Mistral, WizardMATH, OpenMath and DeepSeekMath. Our DSA is significantly better than open-sourced baseline models, even better than ChatGPT models. † means our methods.

| Model | Pre-Calculus | Social Studies |
|---|---|---|
| ChatGPT | 16.1 | 83.5 |
| GPT4 | 29.8 | 95.0 |
| LLAMA2-7b | 0.8 | 53.0 |
| Mistral-7b | 4.6 | 62.0 |
| WizardMATH-7b | 2.5 | 28.5 |
| WizardMATH-v1.1-7b | 16.5 | 68.5 |
| OpenMath-7b | 12.0 | 46.8 |
| DeepSeekMATH-Inst-7b | 16.8 | 66.5 |
| DSA-LLAMA2-7b † | 16.5 | 72.8 |
| DSA-Mistral-7b † | **18.6** | **75.8** |
| LLAMA2-13b | 1.1 | 58.9 |
| WizardMATH-13b | 4.0 | 34.4 |
| DSA-LLAMA2-13b † | **18.8** | **78.0** |
| LLAMA2-70b | 2.6 | 76.2 |
| WizardMATH-70b | 6.9 | 40.6 |
| DSA-LLAMA2-70b † | **22.6** | **87.9** |

Table 3: Accuracy on Pre-Calculus evaluation sets after training LLAMA2-7/13/70b models on textbook descriptions with different training sequences.

| Training Sequence | LLAMA2-7b | LLAMA2-13b | LLAMA2-70b |
|---|---|---|---|
| - | 0.8 | 1.1 | 2.6 |
| Lower to Higher | **8.2** | **9.8** | **14.6** |
| Higher to Lower | 3.2 | 5.8 | 6.2 |
| Random Order 1 | 3.5 | 6.0 | 9.2 |
| Random Order 2 | 3.0 | 4.2 | 5.8 |
| Random Order 3 | 4.8 | 6.8 | 9.5 |

With Dynamic Skill Adaptation framework, we could automatically generate textbook descriptions for skills in the decomposed skill graph and arrange them following the human learning pathways, which allow model to better grasp the knowledge in specialized domains. Furthermore, the exercise which explicitly utilize the decomposed skills together with the dynamic training process empowers LLMs with the ability to better solve the complex problems (e.g., a 304% performance improvement of our DSA-Mistral-7b over general models like Mistral-7b and a 10.7% performance improvement of our DSA-Mistral-7b over specialized models like DeepSeekMATH which leverages wider ranges of human-written corpus).

We further visualize the accuracy on Pre-Calculus evaluation set of every intermediate step when learning textbooks with LLAMA2-7b with *Lower to Higher* orders and the reversed *Higher to Lower* orders in Figure 2 to better illustrate the effectiveness of training following the orders in our constructed skill graph. When accumulating skills following the skill graphs, the blue line (left to right) demonstrates steady performance improvements after learning different level of skills and achieve better final performance compared to the orange line (right to left) which learns the textbook in a reversed order.

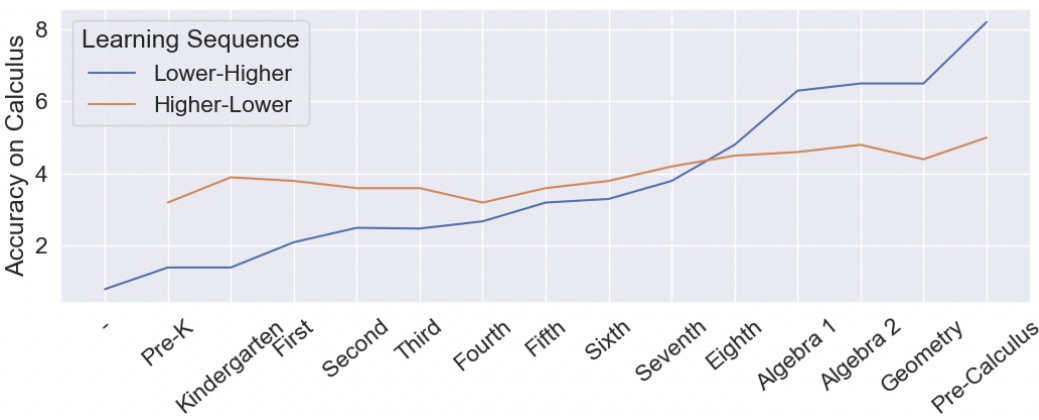

Figure 2: The accuracy on Pre-Calculus evaluation set of every intermediate step when learning textbooks with LLAMA2-7b. The blue line (left to right) represents the process where we arrange the learning sequence following the constructed skill graph from lower-levels to higher-levels while the orange line (right to left) represents the process where the model is learning the textbook in a reversed order.

Table 4: Accuracy on Pre-Calculus and Social Studies when we gradually add each components to LLAMA2-7b models. Note that the last row contains all the components in our DSA framework including textbook descriptions for pre-training, skill graphs to arrange the training sequence, exercise-data for instruction-tuning and dynamic training to update the training data.

| Model | Pre-Calculus | Social Studies |
|---|---|---|
| LLAMA2-7b | 0.8 | 53.0 |
| + textbook | 5.2 | 63.5 |
| + textbook,skill graph | 8.2 | 68.0 |
| + textbook,skill graph,exercise | 12.4 | 70.5 |
| + textbook,skill graph,exercise,dynamic training | **16.5** | **72.8** |

### 4.3 ABLATION STUDIES

To further illustrate the effectiveness of our proposed DSA framework, we perform a set of ablation studies shown below.

**Shuffling the Training Sequence**    We first perform ablation studies on the skill graphs. Specifically, we compare LLAMA2 models which learns the generated textbook descriptions (without exercise instruction-tuning and dynamic training) in different orders: (i)*Lower to Higher* which follows the orders in our constructed skill graph, (ii) *Higher to Lower* which follows a reversed order and (iii) *Random Order 1/2/3*, where we randomly shuffle the constructed skill graph and arrange the training data following the random skill graph. The results on Pre-Calculus with LLAMA2-7/13/70b are shown in Table 3. Reversing orders or corrupting the skill graph would both decrease the performances, suggesting the importance of constructing the skill graph and learning the knowledge following the dependence orders in skill graphs.

**Removing Each Component**    We the perform ablation studies to illustrate the contribution of each component in our DSA framework by gradually adding different component (textbook descriptions for pre-training, arranging the textbook with the skill graph orders, exercise for instruction-tuning and dynamic training) to baseline model (LLAMA2-7b). The results are displayed in Table 4. After training with domain specific textbook descriptions, the performances on Calculus and Social Studies both improve compared to base model. Through learning all the content following the orders in skill

Table 5: Accuracy on Pre-Calculus, MATH, GSM8K and Arithmetic evaluation sets. We directly evaluate our DSA models which are learned for solving calculus problems on other general math evaluation sets. † means our methods.

| Model | Pre-Calculus | MATH | GSM8K | Arithmetic |
|---|---|---|---|---|
| ChatGPT | 16.1 | 36.5 | 82.8 | 88.0 |
| LLAMA2-7b | 0.8 | 2.8 | 12.3 | 20.0 |
| Mistral-7b | 4.6 | 9.1 | 37.8 | 33.5 |
| WizardMATH-7b | 2.8 | 10.7 | 54.9 | 42.0 |
| WizardMATH-v1.1-7b | 16.5 | 33.0 | 83.2 | 48.0 |
| OpenMath-7b | 12.0 | 40.5 | 80.2 | 52.5 |
| DeepSeekMATH-Inst-7b | 16.8 | **46.8** | 82.9 | 52.0 |
| DSA-LLAMA2-7b † | 16.5 | 37.6 | 70.8 | 52.0 |
| DSA-Mistral-7b † | **18.6** | 43.5 | **83.8** | **58.0** |

graph, there are significant performance boosts (e.g., a 57.6% improvement on Pre-Calculus). After instruction-tuning and dynamically update the training set, DSA achieves the best performances on both Pre-Calculus and Social Studies. These demonstrate the effectiveness of every design component in our DSA framework.

**Generalization** We then test the generalization abilities of models which are adapted to solve calculus problems on general math evaluation sets including GSM8K (Cobbe et al., 2021), MATH(Hendrycks et al., 2021) and a constructed arithmetic task where we define 200 new mathematical operations in the problem descriptions and show the results in Table 5. Even though our DSA models are targeted at learning Calculus from the skill graph which decomposes Calculus skills, DSA well generalizes to MATH, GSM8K and Arithmetic tasks compared to baseline models which learn with wider ranges of general math corpus.

## 5 CONCLUSION

In this work, we propose Dynamic Skill Adaptation (DSA) framework to adapt LLMs with novel and complex skills. DSA first decomposes the complex skills and constructing a skill graph, then automatically generates the textbook and exercise for every skill in skill graph and arrange them in a lower-to-higher level orders following the skill graph. Furthermore, DSA dynamically updates the training data during training to avoid overfitting easy-to-learn and error examples. Extensive experiments and ablation studies demonstrate the effectiveness of our proposed DSA. In this work, we only use Calculus and Social Studies as case studies of our DS. For future work, we are interested in expanding to a wider range of domains and merging different experts which are equipped with specialized skills.

## 6 LIMITATION

In this paper, we mainly perform experiments on math and social studies due to the limit of computational resources. However, DSA can be general to other domains because complex skills can also be decomposed based on human prior or LLMs like ChatGPT to construct the skill graphs, with which we could further generate and organize the initial training data. In the future work, we would like to explore more domains. In this work, we limit the range of textbooks till US high schools. However, we think a wider range (e.g., college-level) would bring in more performance gains. In terms of data leakage risks, in our evaluation, we designed one artificial task to avoid the impact of potential data leakage in Table 5 (Arithmetic task) where we randomly design and define mathematical operations which are less likely to be seen by GPT-4. Also, even GPT-4 can not achieve high scores on the pre-calculus evaluation (29.8%) which indicates that the data is less likely to be contaminated. Furthermore, we ran a sanity check about the exact match between testing samples and training data and we did not find any exact match. In the future work, we would include the data contamination assessment (Golchin & Surdeanu, 2023) to avoid the data leakage risks.

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

## A  APPENDIX: TABLES, FIGURES, AND ALGORITHM

---

**Algorithm 1** Dynamic Skill Adaptation

---

**Input** A complex skill $S$, an LLM $F$.
**Output** LLM $F$ that adapts skill $S$.
 1: Build skill graph $G$ that decompose $S$
 2: Textbook corpus = $T$, Exercise corpus = $E$
 3: **for** every skill $s$ in $G$ **do**
 4:     Generate textbook descriptions $t$ for $s$: $T = T \cup (t)$
 5:     Generate exercise $e$ explicitly utilizing $s$: $E = E \cup (e)$
 6: **end for**
 7: **for** every skill $s$ following the order in $G$ **do**
 8:     Fetch textbook data $t \in T$
 9:     Pre-train $F$ with $t$
10: **end for**
11: **while** No convergence **do**
12:     Instruction-tune $F$ with $E$
13:     Compute loss and variance
14:     Categorize $E$ into $E_{easy}, E_{hard}, E_{error}, L_{ambiguous}$
15:     Generate more data $E'_{hard}$ similar to $E_{hard}$
16:     Compose data in $E_{easy}$ to more complex data $E'_{easy}$
17:     $E = E_{ambiguous} \cup E'_{hard} \cup E'_{easy}$
18: **end while**
      return $F$

---

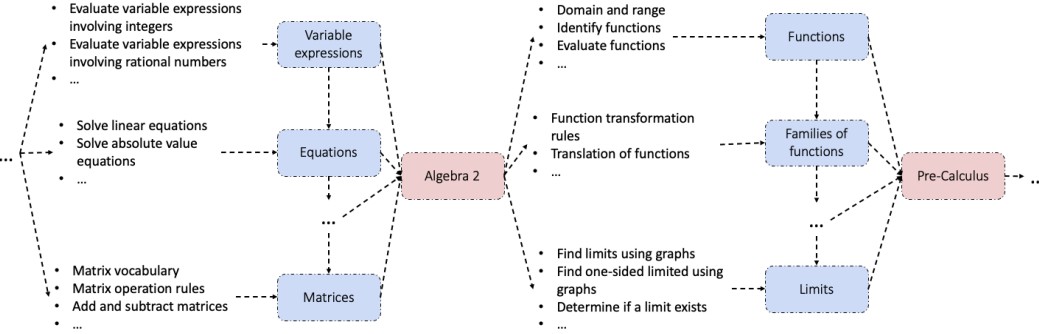

Figure 3: A sub-skill graph in our constructed Calculus skill graph.

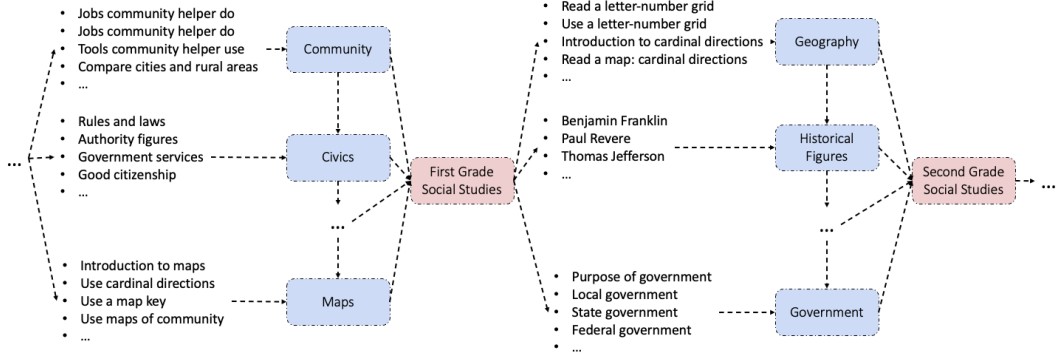

Figure 4: A sub-skill graph in our constructed Social Study skill graph.

Chapter N: Mixed Operations - Addition and Subtraction: Ways to Make a Number

Introduction:
In this chapter, we will explore the concept of addition and subtraction and how they can be used to make a given number.
We will learn different strategies to find combinations of numbers that add up to a given value.
By the end of this chapter, you will be able to confidently use addition and subtraction to make a number.

Section 1: Addition
1.1 Understanding Addition:
Addition is the process of combining two or more numbers to find the total. When we add numbers, the result is called the sum.
For example, if we add 2 and 3, the sum is 5.

1.2 Ways to Make a Number:
To find different ways to make a number, we can use addition. Let's take the number 6 as an example. We can find different
combinations of numbers that add up to 6. Here are a few examples:

Example 1:
6 = 1 + 5
In this example, we added 1 and 5 to get 6.
Example 2:
6 = 2 + 4
Here, we added 2 and 4 to make 6.
Example 3:
6 = 3 + 3
In this case, we added 3 and 3 to get 6.

Section 2: Subtraction
2.1 Understanding Subtraction:
Subtraction is the process of taking away one number from another to find the difference. The result of subtraction is called the
remainder. For example, if we subtract 3 from 7, the remainder is 4.

2.2 Ways to Make a Number:
Similar to addition, we can also use subtraction to find different ways to make a number. Let's continue with the number 6 and explore
some examples:

Example 1:
6 = 7 - 1
In this example, we subtracted 1 from 7 to get 6.
Example 2:
6 = 8 - 2
Here, we subtracted 2 from 8 to make 6.
Example 3:
6 = 9 - 3
In this case, we subtracted 3 from 9 to get 6.

Section 3: Mixed Operations
3.1 Combining Addition and Subtraction:
Now, let's combine addition and subtraction to find different ways to make a number. We will use the number 10 as an example.

Example 1:
10 = 5 + 5
In this example, we added 5 and 5 to get 10.

Example 2:
10 = 12 - 2
Here, we subtracted 2 from 12 to make 10.

Example 3:
10 = 7 + 3
In this case, we added 7 and 3 to get 10.

Exercise:
Now it's time for you to practice! Find different ways to make the number 8 using addition and subtraction. Write down at least
three different combinations.

Solution:
Here are three possible combinations to make the number 8:

8 = 4 + 4
8 = 10 - 2
8 = 6 + 2

Conclusion:
In this chapter, we learned about addition and subtraction and how they can be used to make a given number. We explored different
strategies to find combinations of numbers that add up to a specific value. By practicing these concepts, you will become more
confident in using addition and subtraction to solve problems. Keep up the good work!

Table 6: An example of the generated textbook description.

Four years ago, Kody was only half as old as Mohamed. If Mohamed is currently twice 30 years old, how old is Kody?
Answer:
1. Mohamed is currently twice 30 years old. Using the Skill <Multiplication>, Mohamed is currently 30*2 = 60 years old.
2. Using Skill <Age>, four years ago, Mohamed was 4 years younger than now. Using the Skill <Subtraction>, Mohamed was 60-4 = 56 years old.
3. Four years ago, Kody was only half as old as Mohamed. Using the skill <Division>, Kody was 56/2 = 28 years old.
4. Using Skill <Age>, currently, Kody is 4 years older than four years ago. Using the Skill <Addition>, Kody is currently 28+4 = 32 years old.
5. The answer is 32.

Table 7: An example of the generated exercise.

The definition of economics is:

A. a part of social studies that looks at the way government works
B. a part of social studies that looks at how we meet our wants and needs
C. a part of social studies that looks at how people make important decisions

Table 8: An example of the social study evaluation set.

There is a new mathematical procedure represented as $**$.
The rule of $**$ operation is, for two input numbers a and b,
the output is generated by adding them and the decreasing the
sum by 2. For example, $2 ** 6 = 6$.

Now answer the following question:
What is $12 ** 8$ ?

Table 9: An example of the arithmetic evaluation set.

