# OpenReview forum: "Dynamic Skill Adaptation for Large Language Models"
_ICLR.cc/2025/Conference — Submitted to ICLR 2025_

### Official Review · Reviewer_JgwU · 2024-10-28

**Soundness:** 2
**Presentation:** 2
**Contribution:** 2
**Rating:** 3
**Confidence:** 3

**Summary:**

This work presents a dynamic framework named Dynamic Skill Adaptation (DSA) for LLMs to learn skills regarding problem solving. Specifically, it proposes to first automatically generate and organize the training data by mimicking the learning pathways of human to generate specific skills and exercises and then dynamically tailor the training data based on the training dynamics. Evaluation is conducted by comparing with other LLMs in math reasoning skills and social study skills. Ablation studies are used to show the importance of each module in this framework.

**Strengths:**

S1. The motivation is clear and this paper is also well-structured.

S2. The framework incorporates the human skill learning process and is intuitively insightful.

S3. The evaluation is conducted in several datasets (Pre-Calculus, MATH, GSM8K, Arithmetic, Social Studies) and different tasks (math reasoning and social study).

S4. Ablation studies are also conducted to reveal the importance of each module as well as the training sequence shuffling effect.

**Weaknesses:**

W1. One of my main concerns is the novelty of this work, compared with [1]. From the technical level, this works seem to be an extension of [1] and has marginal novelty compared with [1].

W2. Another concern is that the baseline comparison is not fair. If you compare a fine-tuned model with an old model which is not fine-tuned, then it is not surprising that this fine-tuned model can work better. A real baseline should also fine-tune the model in the same training dataset but without using the fine-tuning strategy in this work. Such baseline is more convincing to show the advantage of this dynamic training scheme in this work.

W3. Generalization abilities. This framework relies on the skill graph construction. Such skills are decomposed by the GPT4. One question is that, if the GPT4 cannot perform well to decompose skills, will the trained LLMs’ performance also drop as well? And does that also mean that the performance of trained LLMs is limited by the upper bound of GPT4? In this case, since we have GPT4, we can just use GPT4 to solve the math questions, which can already achieve very good performance. Why do we need to use GPT4 to first generate skills and train smaller LLMs in order to achieve comparable or even worse performance compared with GPT4? (See table 2, GPT4 has the best performance across all models).

W4. Important details are missing. How do you define the easy-to-learn and hard-to-learn parts? The boundary between them is quite blurred. There is also no detailed information (such as statistics, sample number) regarding the datasets used for either training or evaluation. As such, it is uncertain whether the evaluation is reliable. Moreover, when removing each component in the ablation study, it is unclear whether the model is still fine-tuned. Details of the data used for model training/fine-tuning are also missing.

Reference

[1]. Skill-it! A data-driven skills framework for understanding and training language models, Neurips 2023

**Questions:**

All questions are listed in the weakness above.

**Details Of Ethics Concerns:**

Not sure whether the training data collection process has specific ethic approval.

---

### Official Review · Reviewer_QYed · 2024-10-28

**Soundness:** 2
**Presentation:** 3
**Contribution:** 2
**Rating:** 3
**Confidence:** 4

**Summary:**

This paper introduces Dynamic Skill Adaptation (DSA), a framework that enables large language models (LLMs) to adapt to new and complex skills more effectively. Drawing inspiration from human teaching methodologies, DSA begins by creating a skill graph that breaks down complex skills into sub-skills, organizing them according to their dependencies. For each skill, DSA generates textbook-like explanations and exercise-style data, allowing LLMs to simulate human-like learning paths. Throughout the training, DSA dynamically fine-tunes the training data by reducing the emphasis on simpler examples, generating more challenging instances, and filtering out erroneous data. Experiments with models such as LLAMA and Mistral demonstrate DSA’s effectiveness in enhancing adaptation for tasks like math reasoning and social studies.

**Strengths:**

* The writing is clear and easy to understand.
* Detailed ablation studies are provided to validate the impact of each component.
* The dynamic training approach appears novel, offering a simple yet effective method for data filtering.

**Weaknesses:**

* All data collection steps involve using prompts for GPT-4; however, details on prompt design are lacking. I believe the prompt design could significantly impact performance, as the proposed method heavily depends on data collection.
* The method seems to implicitly distill domain-specific knowledge from GPT-4, but its performance still falls short of the teacher model (GPT-4). This raises a concern: why invest in costly API calls for data collection to create a specialized LLM that performs below the more general GPT-4? A more compelling approach might involve using Llama 2 to generate the data, thereby demonstrating that DSA can effectively improve LLMs' own domain-specific performance.
* Given the broad training data used by GPT-4, a more rigorous analysis of potential data leakage is warranted. Although the authors state that they conducted a sanity check to rule out exact matches between the test samples and training data, it is unlikely for GPT-4 to exactly replicate its training data verbatim. A more convincing approach would be to demonstrate that the exercises generated by GPT-4 do not yield the same answers as any questions in the test set. Besides, DSA models underperform compared to ChatGPT on the authors' custom Arithmetic task, yet outperform on the main tasks (pre-calculus and social studies). This discrepancy raises further concerns that data leakage may be influencing results, with DSA models potentially gaining an advantage by simply memorizing test data.

**Questions:**

Q1: What are the proportions of $E_{easy}$,  $E_{hard}$,  $E_{error}$ and $E_{ambiguous}$ in each iteration? Is the initial error rate high or low? Is the majority of the data ambiguous, or can most be classified into a specific category? I would like to know more about how they change through the iterations. Additionally, there is a minor typo in Algorithm 1: $L_{ambiguous}$ should be $E_{ambiguous}$.

My main concerns lie in the weaknesses outlined above. These issues are significant, and without thorough clarification and detailed analysis of these points, it is challenging to assign a positive score.

---

### Official Review · Reviewer_abLx · 2024-10-31

**Soundness:** 3
**Presentation:** 3
**Contribution:** 2
**Rating:** 6
**Confidence:** 4

**Summary:**

This paper presents Dynamic Skill Adaptation (DSA), a framework for adapting LLMs to acquire novel and complex skills. The approach involves automatically generating training data organized in a skill graph structure inspired by human learning pathways, and dynamically adjusting the training data based on model performance. The authors evaluate their method on math reasoning and social study skills using Llama and Mistral models, claiming improved performance over baselines.

**Strengths:**

- The paper addresses an important problem in LLM adaptation and specialization
- The approach of imitating human learning pathways through skill graphs is intuitive
- The paper is well written and easy to follow

**Weaknesses:**

See the questions below.

**Questions:**

- The proposed hierarchical skill graph assumes a clear, directional relationship between skills. However, many real-world skills exhibit cyclical dependencies (e.g., skill A helps learn B, which in turn reinforces A).The current method's strict hierarchical organization may constrain such bidirectional learning relationships. How does DSA handle these cyclical dependencies?
- How does DSA verify that the model is truly learning the intended skill rather than finding shortcuts? To say it more clearly, how to verify that DSA truly understands the underlying problem structure instead of learns some fixed patterns?
- The method's fundamental assumption that complex skills can be decomposed into clear subtasks may not hold for many important capabilities. Some skills are inherently holistic or emerge from the complex interaction of multiple components that cannot be cleanly separated. How does DSA handle such complex cases?
- The paper's training process focuses on generating harder examples for difficult cases, but does not address how to identify and correct fundamental misunderstandings. Could DSA detect when a model has learned an incorrect approach and actively guide it toward unlearning these mistakes? This seems especially critical given that the method relies on self-generated training data.

---

### Meta-Review · Area_Chair_GKQW · 2024-12-23

**Metareview:**

This paper proposes an interesting method for using language models to learn skill hierarchies. While this is an important problem and the paper makes a novel contribution and is presented clearly, reviewers raised significant concerns regarding novelty, applicability, evaluation, and omitted details, that remain unaddressed.

**Additional Comments On Reviewer Discussion:**

The authors declined to address the reviewers' concerns.

---

### Decision · Program_Chairs · 2025-01-22

Reject